# ON UNDERSTANDING KNOWLEDGE GRAPH REPRESENTATION

## ABSTRACT

Many methods have been developed to represent knowledge graph data, which implicitly exploit low-rank latent structure in the data to encode known information and enable unknown facts to be inferred. To predict whether a relationship holds between entities, their embeddings are typically compared in the latent space following a relation-specific mapping. Whilst link prediction has steadily improved, the latent structure, and hence why such models capture semantic information, remains unexplained. We build on recent theoretical interpretation of word embeddings to derive an explicit structure for the representations of relations between entities. From this we explain properties and justify the relative performance of leading knowledge graph representation methods for identifiable relation types, including their often overlooked ability to make independent predictions.

## 1 INTRODUCTION

Knowledge graphs are large repositories of binary relations between words (or entities) in the form of fact triples *(subject, relation, object)*. Many models have been developed for learning representations of entities and relations in knowledge graphs, such that known facts can be recalled and previously unknown facts can be inferred, a task known as *link prediction*. Recent link prediction models (e.g. Bordes et al., 2013; Trouillon et al., 2016; Balažević et al., 2019b) learn entity representations, or *embeddings*, of far lower dimensionality than the number of entities, by capturing latent structure in the data. Relations are typically represented as a mapping from the embedding of a subject entity to its related object entity embedding(s). Although the performance of knowledge graphlink prediction models has steadily improved for nearly a decade, relatively little is understood of the low-rank latent structure that underpins these models, which we address in this work.

We start by drawing a parallel between entity embeddings in knowledge graphs and unsupervised word embeddings, as learned by algorithms such as Word2Vec (W2V) (Mikolov et al., 2013) and GloVe (Pennington et al., 2014). We assume that words have latent features, e.g. meaning(s), tense, grammatical type, that are innate and fixed, irrespective of what an embedding may capture (which may be only a part, subject to the embedding method and/or the data source); and that this same latent structure gives rise to patterns observed in the data, e.g. in word co-occurrence statistics and in which words are related to which. As such, an understanding of the latent structure from one embedding task (e.g. word embedding) might be useful to another (e.g. knowledge graph entity embedding).

Recent work (Allen & Hospedales, 2019; Allen et al., 2019) theoretically explains how semantic properties are encoded in word embeddings that (approximately) factorise a matrix of word co-occurrence *pointwise mutual information* (PMI), e.g. as is known for W2V (Levy & Goldberg, 2014). Semantic relationships between words (specifically similarity, relatedness, paraphrase and analogy) are proven to manifest as linear relationships between rows of the PMI matrix (subject to known error terms), of which word embeddings can be considered low-rank projections. This explains why similar words (e.g. synonyms) have similar embeddings; and embeddings of analogous word pairs share a common "vector offset".

Importantly, this insight allows us to identify geometric relationships between such word embeddings necessary for other semantic relations to hold, such as those of knowledge graphs. These *relation conditions* describe relation-specific mappings between entity embeddings, i.e. relation representations, providing a "blue-print" against which to consider knowledge graph representation models. We find that various properties of knowledge graph representation models, including the relative

**Table 1:** Score functions of representative linear link prediction models. $\boldsymbol{R} \in \mathbb{R}^{d_e \times d_e}$ and $\boldsymbol{r} \in \mathbb{R}^d_e$ are the relation matrix and translation vector, $\mathbf{W} \in \mathbb{R}^{d_e \times d_r \times d_e}$ is the core tensor and $b_s, b_o \in \mathbb{R}$ are the entity biases.

| Model | | Linear Subcategory | Score Function |
|---|---|---|---|
| TransE | (Bordes et al., 2013) | additive | $-\|\boldsymbol{e}_s + \boldsymbol{r} - \boldsymbol{e}_o\|_2^2$ |
| DistMult | (Yang et al., 2015) | multiplicative (diagonal) | $\boldsymbol{e}_s^\top \boldsymbol{R} \boldsymbol{e}_o$ |
| TuckER | (Balažević et al., 2019b) | multiplicative | $\mathbf{W} \times_1 \boldsymbol{e}_s \times_2 \boldsymbol{r} \times_3 \boldsymbol{e}_o$ |
| MuRE | (Balažević et al., 2019a) | multiplicative (diagonal) + additive | $-\|\boldsymbol{R}\boldsymbol{e}_s + \boldsymbol{r} - \boldsymbol{e}_o\|_2^2 + b_s + b_o$ |

performance of leading link prediction models, accord with predictions based on these relation conditions, suggesting a commonality to the latent structure learned in word embedding models and knowledge graph representation models, despite the significant differences between their training data and methodology. In summary, the key contributions of this work are:

- to use recent understanding of PMI-based word embeddings to derive what a relation representation must achieve to map a subject word embedding to all related object word embeddings (relation conditions), based on which relations can be categorised into three *types*;
- to show that properties of knowledge graph models fit predictions made from relation conditions, e.g. strength of a relation's *relatedness* aspect is reflected in the eigenvalues of its relation matrix;
- to show that the performance per relation of leading link prediction models corresponds to the ability of the model's architecture to meet the relation conditions of the relation's type, i.e. the better the architecture of a knowledge graph representation model aligns with the form theoretically derived for PMI-based word embeddings, the better the model performs; and
- noting how ranking metrics can be flawed, to provide novel insight into the prediction accuracy per relation of recent knowledge graph models, an evaluation metric we recommend in future.

## 2 BACKGROUND

Our work draws on knowledge graph representation and word embedding. Whilst related, these tasks differ materially in their training data. The former is restricted to datasets crafted by hand or automatically generated, the latter has the vast abundance of natural language text (e.g. Wikipedia).

### 2.1 KNOWLEDGE GRAPH REPRESENTATION

Almost all recent knowledge graph models represent entities $e_s, e_o$ as vectors $\boldsymbol{e}_s, \boldsymbol{e}_o \in \mathbb{R}^{d_e}$ of low dimension (e.g. $d_e = 200$) relative to the number of entities $n_e$ (typically of order $10^4$), and relations as transformations in the latent space from subject entity embedding to object. These models are distinguished by their *score function*, which defines (i) the form of the relation transformation, e.g. matrix multiplication, vector addition; and (ii) how "closeness" between the transformed subject embedding and an object embedding is evaluated, e.g. dot product, Euclidean distance. Score functions can be non-linear (e.g. Dettmers et al. (2018)), or linear and sub-categorised as additive, multiplicative or both. We focus on linear models due to their simplicity and strong performance at link prediction (including state-of-the-art). Table 1 shows the score functions of competitive linear models that span all linear sub-categories: TransE (Bordes et al., 2013), DistMult (Yang et al., 2015), TuckER (Balažević et al., 2019b) and MuRE (Balažević et al., 2019a).

**Additive models** typically use Euclidean distance and contain a relation-specific *translation* from a (possibly transformed) subject to a (possibly transformed) object entity embedding. A generic additive score function is given by $\phi(e_s, r, e_o) = -\|\boldsymbol{R}_s \boldsymbol{e}_s + \boldsymbol{r} - \boldsymbol{R}_o \boldsymbol{e}_o\|_2^2 + b_s + b_o = -\|\boldsymbol{e}_s^{(r)} + \boldsymbol{r} - \boldsymbol{e}_o^{(r)}\|_2^2 + b_s + b_o$. The simplest example is TransE for which $\boldsymbol{R}_s = \boldsymbol{R}_o = \boldsymbol{I}$ and $b_s = b_o = 0$. The score function of MuRE has $\boldsymbol{R}_o = \boldsymbol{I}$ and so combines multiplicative ($\boldsymbol{R}_s = \boldsymbol{R}$) and additive ($\boldsymbol{r}$) components.

**Multiplicative models** have the generic score function $\phi(e_s, r, e_o) = \boldsymbol{e}_s^\top \boldsymbol{R} \boldsymbol{e}_o = \langle \boldsymbol{e}_s^{(r)}, \boldsymbol{e}_o \rangle$, i.e. a *bilinear product* of the entity embeddings and a relation-specific matrix $\boldsymbol{R}$. DistMult is a simple example with diagonal $\boldsymbol{R}$ and so cannot model asymmetric relations (Trouillon et al., 2016). In TuckER, each relation-specific $\boldsymbol{R} = \mathbf{W} \times_3 \boldsymbol{r}$ is a linear combination of $d_r$ "prototype" relation matrices in a core tensor $\mathbf{W} \in \mathbb{R}^{d_e \times d_r \times d_e}$ (where $\times_n$ denotes tensor product along mode $n$), facilitating *multi-task learning* across relations.

## 2.2 Word embedding

Algorithms such as Word2Vec (Mikolov et al., 2013) and GloVe (Pennington et al., 2014) generate succinct low-rank word embeddings that perform well on downstream tasks (Baroni et al., 2014). Such models predict the context words ($c_j$) observed around each target word ($w_i$) in a text corpus using shallow neural networks. Whilst recent language models (e.g. Devlin et al. (2018); Peters et al. (2018)) create impressive *context-specific* word embeddings, we focus on the former embeddings since knowledge graph entities have no obvious context and, more importantly, they are *interpretable*.

Levy & Goldberg (2014) show that, for a dictionary of unique words $\mathbb{D}$ and embedding dimension $d \ll |\mathbb{D}|$, W2V's loss function is minimised when its weight matrices $\boldsymbol{W}, \boldsymbol{C} \in \mathbb{R}^{d \times |\mathbb{D}|}$ (whose columns are word embeddings $\boldsymbol{w}_i, \boldsymbol{c}_j$) factorise a word co-occurrence *pointwise mutual information* (PMI) matrix, subject to a shift term ($\mathrm{PMI}(w_i, c_j) = \log \frac{P(w_i, c_j)}{P(w_i)P(c_j)}$). This connects W2V to earlier count-based embeddings and specifically to PMI, which has a long history in linguistic analysis (Turney & Pantel, 2010). From its loss function, GloVe can be seen to perform a comparable factorisation.

Recent work shows why word embeddings that factorise such PMI matrix encode semantic word relationships (Allen & Hospedales, 2019; Allen et al., 2019). The authors show that word embeddings can be seen as low-rank projections of high dimensional *PMI vectors* (rows of the PMI matrix), between which the semantic relationships of similarity, relatedness, paraphrase and analogy provably manifest as linear geometric relationships (subject to defined error terms), which are then preserved, under a sufficiently linear projection, between word embeddings. Thus *similar* words have similar embeddings, and the embeddings of *analogous* word pairs share a common vector offset.

Specifically, the PMI vectors ($\boldsymbol{p}_x$) of an analogy "*man* is to *king* as *woman* is to *queen*" satisfy $\boldsymbol{p}_Q - \boldsymbol{p}_W \approx \boldsymbol{p}_K - \boldsymbol{p}_M$ because the difference between words associated with *king* and *man* (e.g. *reign*, *crown*) mirrors that between *queen* and *woman*. This leads to a common difference between their co-occurrence distributions (over all words), giving a common difference between their PMI vectors, which projects to a common difference between embeddings. Any discrepancy in the mirroring of word associations is shown to introduce error, weakening the analogy, as does a lack of statistical independence within certain word pairs (see (Allen & Hospedales, 2019)). The common difference in word co-occurrence distributions, e.g. the increased association with words {*reign*, *crown*, etc.}, can be interpreted semantically as a common change in context (*context-shift*) that transforms *man* to *king* and *woman* to *queen* by adding a *royal* context. Under this interpretation, context can also be subtracted, e.g. "*king* is to *man* as *queen* is to *woman*" (minus *royal*); or both, e.g. "*boy* is to *king* as *girl* is to *queen*" (minus *youth* plus *royal*). Adding context can also be interpreted as *specialisation*, and subtracting context as *generalisation*. This establishes a correspondence between common word embedding vector offsets and semantic context-shifts.

Although the projection from PMI vectors to word embeddings preserves the *relative* relationships, and thus the above semantic interpretability of common embedding differences, a direct interpretation of dimensions themselves is obscured, not least because any embedding matrix can be arbitrarily scaled/rotated if the other is inversely transformed.

## 3 Relationships between embeddings of related words

Our aim is to build on the understanding of PMI-based word embeddings (henceforth *word embeddings*), to identify what a knowledge graph relation representation needs to achieve to map all subject word embeddings to all related object word embeddings. We note that if a semantic relation between two words implies a particular geometric relationship between their embeddings, then the latter serves as a necessary quantitative condition for the former to hold (a *relation condition*). Relation conditions implicitly define a relation-specific mapping by which all subject embeddings are mapped to all related object embedding(s), allowing related entities to be identified by a proximity measure (e.g. Euclidean distance or dot product). Since this is the approach of many knowledge graph representation models, their performance can be contrasted with their ability to express mappings that satisfy required relation conditions. For example, similarity and context-shift relations respectively imply closeness and a relation-specific vector offset between embeddings (S2.2). Such relation conditions can be tested for by respectively making no change to the subject entity or adding the relation-specific offset, before measuring proximity with the object. We note that since relation conditions are not necessarily sufficient, they do not guarantee a relation holds, i.e. false positives may arise.

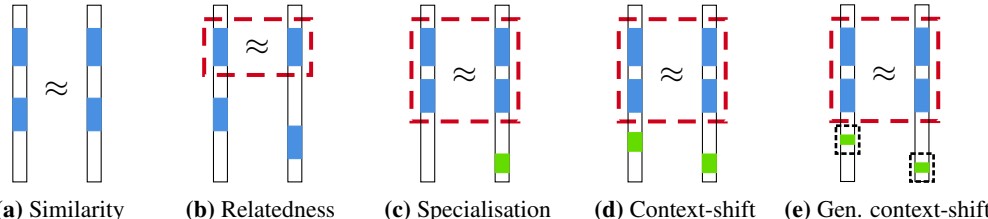

**(a)** Similarity     **(b)** Relatedness     **(c)** Specialisation     **(d)** Context-shift     **(e)** Gen. context-shift

**Figure 1:** Relationships between PMI vectors of entities under different relation types. Shaded regions indicate strong word associations (positive PMI values); red lines indicate relatedness; black lines denote context sets.

In general, the data from which knowledge graph embeddings are derived differs greatly to the co-occurrence data used for word embeddings, and the latter would not be anticipated to be learned by knowledge graph models. However, word embeddings provide a *known solution* (i.e. minimise the loss function) of any knowledge graph model able to express the required mapping(s) derived from relation conditions, where the loss function measures proximity between mapped entities.

The relation conditions for certain relation types (underlined) follow readily from S2.2:

• **Similarity:** Semantically similar words induce similar distributions over the words they co-occur with. Thus their PMI vectors (Fig 1a) and word embeddings are similar.

• **Relatedness:** The relatedness of two words can be defined in terms of the words with which both co-occur similarly ($\mathbb{S} \in \mathbb{D}$), which define the *nature* of relatedness, e.g. *milk* and *cheese* are related by $\mathbb{S} = \{dairy, breakfast, ...\}$; and $|\mathbb{S}|$ reflects the *strength* of relatedness. Since PMI vector components corresponding to $\mathbb{S}$ are similar (Fig 1b), embeddings of "$\mathbb{S}$-*related*" words have similar components in the subspace $\mathbb{V}_\mathbb{S}$ that spans the projected PMI vector dimensions corresponding to $\mathbb{S}$. The rank of $\mathbb{V}_\mathbb{S}$ might be expected to reflect relatedness strength. In general, relatedness is a weaker, more variable relation than similarity, its limiting case with $\mathbb{S} = \mathbb{D}$ and rank($\mathbb{V}_\mathbb{S}$) $= d$.

• **Context-shift:** In the context of word embeddings, *analogy* typically refers to *relational similarity* (Turney, 2006; Gladkova et al., 2016). More specifically, the relations within analogies that give a common vector offset between word embeddings require related words to have a common difference between their distributions of co-occurring words, defined as a *context-shifts* (see S2.2). These relations are strictly 1-to-1 and include an aspect of *relatedness* due to the word associations in common (Fig 1d). A *specialisation* relation is a context-shift in which context is only added (Fig 1c).

• **Generalised context-shift:** Context-shift relations are generalised to 1-to-many, many-to-1 and many-to-many relations by letting the fully-specified added or subtracted context be one of a (relation-specific) *context set* (Fig 1e), e.g. allowing an entity to be *any* colour or *anything* blue. The potential scope and size of each context set means these relations can vary greatly. The limiting case for small context sets has a single context in each, whereby the relation is an explicit context-shift (as above), and the *difference* between embeddings is a known vector offset. In the limiting case where context sets are large, the added/subtracted context is so loosely defined that, in effect, only the relatedness aspect of the relation and thus only the common subspace component of embeddings is known.

**Link to set theory:** Viewing PMI vectors as *sets of word associations* and taking intuition from Fig 1, the above relations can be seen to reflect set operations: similarity as set equality; relatedness as equality of a subset; and context-shift as the set difference equalling a relation-specific set. This highlights how the relatedness aspect of a relation reflects features that must be common, and context-shift reflects features that must differ. Whilst this mirrors an intuitive notion of "feature vectors", we emphasise that this is grounded in the co-occurrence statistics of PMI-based word embeddings.

## 3.1 CATEGORISING KNOWLEDGE GRAPH RELATIONS

Analysing the relations of popular knowledge graph datasets with the above perspective, we find that they comprise (i) a relatedness aspect that reflects a common theme (e.g. both entities are animals or geographic terms); and (ii) specific word associations of the subject and/or object entities. Specifically, relations appear to fall under a hierarchy of three *relation types*: highly related (**R**); (generalised) specialisation (**S**); and (generalised) context-shift (**C**). As above, "generalised" indicates that any added/subtracted contexts can be from a set. From Fig 1, type R relations can be seen as a special case of S, which, in turn, is a special case of C. Type C is therefore a generalised case of all considered relations. Whilst there are several other ways to classify relations (e.g. by their hierarchy,

**Table 2:** Categorisation of WN18RR relations.

| Type | Relation | Examples *(subject entity, object entity)* |
|---|---|---|
| R | verb_group
derivationally_related_form
also_see | *(trim_down_VB_1, cut_VB_35), (hatch_VB_1, incubate_VB_2)*
*(lodge_VB_4, accommodation_NN_4), (question_NN_1, inquire_VB_1)*
*(clean_JJ_1, tidy_JJ_1), (ram_VB_2, screw_VB_3)* |
| S | hypernym
instance_hypernym | *(land_reform_NN_1, reform_NN_1), (prickle-weed_NN_1, herbaceous_plant_NN_1)*
*(yellowstone_river_NN_1, river_NN_1), (leipzig_NN_1, urban_center_NN_1)* |
| C | member_of_domain_usage
member_of_domain_region
member_meronym
has_part
synset_domain_topic_of | *(colloquialism_NN_1, figure_VB_5), (plural_form_NN_1, authority_NN_2)*
*(rome_NN_1, gladiator_NN_1), (usa_NN_1, multiple_voting_NN_1)*
*(south_NN_2, sunshine_state_NN_1), (genus_carya_NN_1, pecan_tree_NN_1)*
*(aircraft_NN_1, cabin_NN_3), (morocco_NN_1, atlas_mountains_NN_1)*
*(quark_NN_1, physics_NN_1), (harmonize_VB_3, music_NN_4)* |

transitivity), by considering relation conditions, we delineate by the required mathematical form (and complexity) of their representation. Table 2 shows a categorisation of the relations of the WN18RR dataset (Dettmers et al., 2018), containing 11 relations between 40,943 entities.[1] An explanation for this assignment is in Appx. A and that for NELL-995 (Xiong et al., 2017) is in Appx. B. A review of the FB15k-237 dataset (Toutanova et al., 2015) shows the vast majority of relations to be of type C preventing a contrast between relation types being drawn, hence we do not consider that dataset.

### 3.2 Relations as mappings of embeddings

Given the relation conditions of a particular relation type, we can recognise mappings that meet them and thus loss functions (that evaluate the proximity of mapped entity embeddings by dot product or Euclidean distance) able to identify relations of that type *between PMI-based word embeddings*. We then contrast these theoretically inspired loss functions (one per relation type) with those of knowledge graph models (Table 1) and, on the outline assumption that a common low-rank latent structure is exploited by both word embeddings and knowledge graph models, predict properties and the relative performance of different knowledge graphs models for different relation types.

**R:** To evidence $\mathbb{S}$-relatedness, both entity embeddings $\boldsymbol{e}_s, \boldsymbol{e}_o$ must be projected onto a subspace $\mathbb{V}_{\mathbb{S}}$, where their images are compared. Projection requires multiplication by a matrix $\boldsymbol{P}_r \in \mathbb{R}^{d \times d}$ and cannot be achieved additively, except in the limiting case of similarity, when $\boldsymbol{P}_r = \boldsymbol{I}$ or vector $\boldsymbol{r} \approx \boldsymbol{0}$ is added. Comparison by dot product gives $(\boldsymbol{P}_r \boldsymbol{e}_s)^\top (\boldsymbol{P}_r \boldsymbol{e}_o) = \boldsymbol{e}_s^\top \boldsymbol{P}_r^\top \boldsymbol{P}_r \boldsymbol{e}_o = \boldsymbol{e}_s^\top \boldsymbol{M}_r \boldsymbol{e}_o$ (for a relation-specific symmetric $\boldsymbol{M}_r = \boldsymbol{P}_r^\top \boldsymbol{P}_r$). Euclidean distance gives $\|\boldsymbol{P}_r \boldsymbol{e}_s - \boldsymbol{P}_r \boldsymbol{e}_o\|^2 = (\boldsymbol{e}_s - \boldsymbol{e}_o)^\top \boldsymbol{M}_r (\boldsymbol{e}_s - \boldsymbol{e}_o) = \|\boldsymbol{P}_r \boldsymbol{e}_s\|^2 - 2 \boldsymbol{e}_s^\top \boldsymbol{M}_r \boldsymbol{e}_o + \|\boldsymbol{P}_r \boldsymbol{e}_o\|^2$.

**S/C:** Evidencing these relations requires a test both for $\mathbb{S}$-relatedness and for relation-entity-specific embeddings component(s) $(\boldsymbol{v}_r^s, \boldsymbol{v}_r^o)$. This can be achieved by (i) multiplying both entity embeddings by a relation-specific projection matrix $\boldsymbol{P}_r$ that projects onto the subspace that spans the low-rank projection of dimensions corresponding to $\mathbb{S}$, $\boldsymbol{v}_r^s$ and $\boldsymbol{v}_r^o$, (which tests for $\mathbb{S}$-relatedness whilst preserving any entity-specific embedding components); and (ii) adding a relation-specific vector $\boldsymbol{r} = \boldsymbol{v}_r^o - \boldsymbol{v}_r^s$ to the transformed subject entity embeddings. Comparison of the final transformed entity embeddings by dot product equates to $(\boldsymbol{P}_r \boldsymbol{e}_s + \boldsymbol{r})^\top \boldsymbol{P}_r \boldsymbol{e}_o$; and by Euclidean distance to $\|\boldsymbol{P}_r \boldsymbol{e}_s + \boldsymbol{r} - \boldsymbol{P}_r \boldsymbol{e}_o\|^2 = \|\boldsymbol{P}_r \boldsymbol{e}_s + \boldsymbol{r}\|^2 - 2(\boldsymbol{P}_r \boldsymbol{e}_s + \boldsymbol{r})^\top \boldsymbol{P}_r \boldsymbol{e}_o + \|\boldsymbol{P}_r \boldsymbol{e}_o\|^2$ (*cf* MuRE: $\|\boldsymbol{R} \boldsymbol{e}_s + \boldsymbol{r} - \boldsymbol{e}_o\|^2$).

Contrasting the above loss functions with those of knowledge graph models (Table 1), we make the following predictions: (**P1**) the ability to learn the representation of a particular relation is expected to reflect the complexity of its type (R>S>C), and whether all relation conditions (e.g. additive or multiplicative interactions) can be met under a given model; (**P2**) relation matrices for relatedness (type R) relations are highly symmetric; (**P3**) offset vectors for relatedness relations have low norm; and (**P4**) as a proxy to the rank of $\mathbb{V}_{\mathbb{S}}$, the eigenvalues of a relation matrix reflect a relation's strength of relatedness. To elaborate: P1 anticipates that additive-only models (e.g. TransE) are not suited to identifying the relatedness aspect of relations (except in limiting cases of similarity, requiring a zero vector); and multiplicative-only models (e.g. DistMult) should perform well on type R but are not suited to identifying entity-specific features of type S/C, for which an asymmetric relation matrix in TuckER may help compensate. Further, the loss function of MuRE closely resembles that derived for type C relations (which generalise all others) and is thus expected to perform best overall.

---

[1]We omit the relation "similar_to" since its instances have no discernible structure, and only 3 occur in the test set, all of which are the inverse of a training example and trivial to predict.

**Table 3:** Hits@10 per relation on WN18RR.

| Relation Name | Type | % | # | Khs | Max/Avg Path | | TransE | MuRE$_I$ | DistMult | TuckER | MuRE |
|---|---|---|---|---|---|---|---|---|---|---|---|
| verb_group | R | 1% | 39 | 0.00 | 1 | 1.0 | 0.87 | 0.95 | **0.97** | **0.97** | **0.97** |
| derivationally_related_form | R | 34% | 1074 | 0.04 | 1 | 1.0 | 0.93 | 0.96 | 0.96 | 0.96 | **0.97** |
| also_see | R | 2% | 56 | 0.24 | 44 | 15.2 | 0.59 | **0.73** | 0.67 | 0.72 | **0.73** |
| instance_hypernym | S | 4% | 122 | 1.00 | 3 | 1.0 | 0.22 | 0.52 | 0.47 | 0.53 | **0.54** |
| synset_domain_topic_of | C | 4% | 114 | 0.99 | 3 | 1.1 | 0.19 | 0.43 | 0.42 | 0.45 | **0.53** |
| member_of_domain_usage | C | 1% | 24 | 1.00 | 2 | 1.0 | 0.42 | 0.42 | 0.48 | 0.38 | **0.50** |
| member_of_domain_region | C | 1% | 26 | 1.00 | 2 | 1.0 | 0.35 | 0.40 | 0.40 | 0.35 | **0.46** |
| member_meronym | C | 8% | 253 | 1.00 | 10 | 3.9 | 0.04 | 0.38 | 0.30 | **0.39** | 0.39 |
| has_part | C | 6% | 172 | 1.00 | 13 | 2.2 | 0.04 | 0.31 | 0.28 | 0.29 | **0.35** |
| hypernym | S | 40% | 1251 | 0.99 | 18 | 4.5 | 0.02 | 0.20 | 0.19 | 0.20 | **0.28** |
| all | | 100% | 3134 | | | | 0.38 | 0.52 | 0.51 | 0.53 | **0.57** |

**Table 4:** Hits@10 per relation on NELL-995.

| Relation Name | Type | % | # | Khs | Max/Avg Path | | TransE | MuRE$_I$ | DistMult | TuckER | MuRE |
|---|---|---|---|---|---|---|---|---|---|---|---|
| teamplaysagainstteam | R | 2% | 243 | 0.11 | 10 | 3.5 | 0.76 | 0.84 | **0.90** | 0.89 | 0.89 |
| clothingtogowithclothing | R | 1% | 132 | 0.17 | 5 | 2.6 | 0.72 | 0.80 | **0.88** | 0.85 | 0.84 |
| professionistypeofprofession | S | 1% | 143 | 0.38 | 7 | 2.5 | 0.37 | 0.55 | 0.62 | 0.65 | **0.66** |
| animalistypeofanimal | S | 1% | 103 | 0.68 | 9 | 3.1 | 0.50 | 0.56 | 0.64 | **0.68** | 0.65 |
| athleteplayssport | C | 1% | 113 | 1.00 | 1 | 1.0 | 0.54 | 0.58 | 0.58 | 0.60 | **0.64** |
| chemicalistypeofchemical | S | 1% | 115 | 0.53 | 6 | 3.0 | 0.23 | 0.43 | 0.49 | 0.51 | **0.60** |
| itemfoundinroom | C | 2% | 162 | 1.00 | 1 | 1.0 | 0.39 | 0.57 | 0.53 | 0.56 | **0.59** |
| agentcollaborateswithagent | R | 1% | 119 | 0.51 | 14 | 4.7 | 0.44 | 0.58 | **0.64** | 0.61 | 0.58 |
| bodypartcontainsbodypart | C | 1% | 103 | 0.60 | 7 | 3.2 | 0.30 | 0.38 | 0.54 | **0.58** | 0.55 |
| atdate | C | 10% | 967 | 0.99 | 4 | 1.1 | 0.41 | 0.50 | 0.48 | 0.48 | **0.52** |
| locationlocatedwithinlocation | C | 2% | 157 | 1.00 | 6 | 1.9 | 0.35 | 0.37 | 0.46 | **0.48** | 0.48 |
| atlocation | C | 1% | 294 | 0.99 | 6 | 1.4 | 0.22 | 0.33 | 0.39 | 0.43 | **0.44** |
| all | | 100% | 20000 | | | | 0.36 | 0.48 | 0.51 | **0.52** | **0.52** |

## 4 COMPARING KNOWLEDGE GRAPH MODELS

We test the predictions made on the basis of word embeddings by comparing the performance of competitive knowledge graph models, TransE, DistMult, TuckER and MuRE (see S2), which entail different forms of relation representation, on all WN18RR relations and a similar number of NELL-995 relations (selected to represent each relation type). Since applying the logistic sigmoid to the score function of TransE does not give a probabilistic interpretation comparable to other models, we include MuRE$_I$, a constrained variant of MuRE with $\boldsymbol{R}_s = \boldsymbol{R}_o = \boldsymbol{I}$, as a proxy to TransE for a fairer comparison. Implementation details are included in Appx. D. For evaluation, we generate $2n_e$ *evaluation triples* for each test triple (for the number of entities $n_e$) by fixing the subject entity $e_s$ and relation $r$ and replacing the object entity $e_o$ with all possible entities and then keeping $e_o$ and $r$ fixed and varying $e_s$. The obtained scores are ranked to give the standard metric hits@10 (Bordes et al., 2013), i.e. the fraction of times a true triple appears in the top 10 ranked evaluation triples.

### 4.1 PERFORMANCE PER RELATION TYPE

Tables 3 and 4 report results (hits@10) for each relation and include the relation type and known confounding influences: percentage of relation instances in the training and test sets (approximately equal), number of instances in the test set, Krackhardt hierarchy score (see Appx. E) (Krackhardt, 2014; Balažević et al., 2019a) and maximum and average shortest path between any two related nodes. A further confounding effect is dependence between relations. Balažević et al. (2019b) and Lacroix et al. (2018) show that constraining the rank of relation representations benefits datasets with many relations (particularly when the number of instances per relation is low) due to *multi-task learning*, which is expected to benefit TuckER on the NELL-995 dataset (200 relations). Note that all models have a comparable number of free parameters.

**P1:** As predicted, all models tend to perform best at type R relations, with a clear performance gap to other relation types. Also, performance on type S relations appears higher in general than type C. Additive-only models (TransE, MuRE$_I$) perform most poorly on average, in line with prediction since all relation types involve a relatedness component. They achieve their best results on type R relations, where the relation vector can be zero/small. Multiplicative-only DistMult performs well, sometimes best, on type R relations, fitting expectation as it can fully represent those relations and has no additional parameters that may overfit to noise (which may explain where MuRE performs

slightly worse). As expected, MuRE, performs best on average (particularly on WN18RR), and most strongly on S and C type relations that require both multiplicative and additive components. The comparable performance of TuckER on NELL-995 is explained by its ability for multi-task learning.

Other unexpected results also closely align with confounding factors, e.g. that all models perform poorly on the *hypernym* relation, despite it having type S and a relative abundance of training data (40% of all instances), might be explained by its hierarchical nature (Khs ≈ 1 and long paths). The same may explain the reduced performance on type R relations *also_see* and *agentcollaborateswithagent*. As found previously, none of the models considered are well suited to modelling hierarchical structures (Balažević et al., 2019a). We also note that the percentage of training instances of a relation does not seem to correlate with its performance, as might typically be expected.

**P2/P3:** Table 5 shows the symmetry score ($\in$ [-1, 1] indicating perfect anti-symmetry to symmetry; see Appx. F) for the relation matrix of TuckER and the norm of relation vectors of TransE, MuRE$_I$ and MuRE on the WN18RR dataset. As expected, type R relations have high symmetry, whereas both other relation types have lower scores, fitting the expectation that TuckER compensates for having no additive component. All additive models learn relation vectors of a noticeably lower norm for type R relations (where, in the extreme, no additive component is required) than for types S and C.

**Table 5:** Relation matrix symmetry score [-1.1] for TuckER; and relation vector norm for TransE, MuRE$_I$ and MuRE (WN18RR).

| | | Symmetry Score | Vector Norm | | |
|---|---|---|---|---|---|
| Relation | Type | TuckER | TransE | MuRE$_I$ | MuRE |
| verb_group | R | 0.52 | 5.65 | 0.76 | 0.89 |
| derivationally_related_form | R | 0.54 | 2.98 | 0.45 | 0.69 |
| also_see | R | 0.50 | 7.20 | 0.97 | 0.97 |
| instance_hypernym | S | 0.13 | 18.26 | 2.98 | 1.88 |
| member_of_domain_usage | C | 0.10 | 11.24 | 3.18 | 1.88 |
| member_of_domain_region | C | 0.06 | 12.52 | 3.07 | 2.11 |
| synset_domain_topic_of | C | 0.12 | 23.29 | 2.65 | 1.52 |
| member_meronym | C | 0.12 | 4.97 | 1.91 | 1.97 |
| has_part | C | 0.13 | 6.44 | 1.69 | 1.25 |
| hypernym | S | 0.04 | 9.64 | 1.53 | 1.03 |

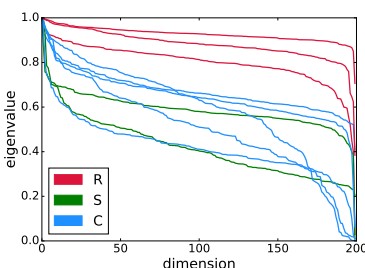

**Figure 2:** Eigenvalue magnitudes of relation-specific matrices $\boldsymbol{R}$ for MuRE by relation type (WN18RR).

**P4:** Fig 2 shows eigenvalue magnitudes (scaled relative to their largest and ordered) of the relation-specific matrices $\boldsymbol{R}$ of MuRE, labelled by relation type. Predicted to reflect strength of a relation's relatedness, they should be highest for type R relations, as observed. For relation types S and C the profiles are more varied, fitting the expectation that the relatedness of those types has greater variability in both choice and size of $\mathbb{S}$, i.e. in the nature and strength of relatedness.

In summary, the results support all predictions made based on the assumption that knowledge graph models benefit from the same latent semantic structure as word embeddings and the relation conditions theoretically derived from them. Our analysis identifies the best performing model per relation type: multiplicative-only DistMult for type R, additive-multiplicative MuRE for types S/C; providing a basis for *dataset-dependent model selection*. The per-relation insight into where models perform poorly, e.g. on hierarchical or type C relations, can also be used to aid and direct future model design.

## 4.2 KNOWLEDGE GRAPH MODEL PREDICTIONS

Even though in practice we want to know whether a particular triple is true or false, such independent predictions are not commonly reported or evaluated. Despite many recent link prediction models being able to independently predict the truth of each triple, it is common practice to report ranking-based metrics, e.g. mean reciprocal rank, hits@$k$, which compare the prediction of a test triple to those of all evaluation triples (see S4). Not only is this computationally costly, the evaluation is flawed if entities are related to more than $k$ others and does not evaluate a model's ability to independently predict whether "$a$ is related to $b$". We address this by considering actual model predictions.

Since for each relation there are $n_e^2$ possible entity-entity relationships, we sub-sample by computing predictions for all $(e_s, r, e_o)$ triples only for each $e_s, r$ pair seen in the test set. We split positive predictions ($\sigma(\phi(e_s, r, e_o)) > 0.5$) between (i) training and test/validation instances (known truths); and (ii) *other*, the truth of which is not known. Per relation, we then compute accuracy over the training instances (train) and the test/validation instances (test); and the average number of other truths predicted per $e_s, r$ pair. Table 6 shows results for MuRE$_I$, DistMult, TuckER and MuRE. All

**Table 6:** Per relation prediction accuracy for MuRE$_I$ (M$_I$), (D)istMult, (T)uckER and (M)uRE (WN18RR).

| Relation Name | $\#_{train}$ | $\#_{test}$ | Accuracy (train) | | | | Accuracy (test) | | | | # Other "True" | | | |
|---|---|---|---|---|---|---|---|---|---|---|---|---|---|---|
| | | | M$_I$ | D | T | M | M$_I$ | D | T | M | M$_I$ | D | T | M |
| verb_group | 15 | 39 | 1.00 | 1.00 | 1.00 | 1.00 | 0.97 | 0.97 | 0.97 | 0.97 | 8.3 | 1.7 | 0.9 | 2.7 |
| derivationally_related_form | 1714 | 1127 | 1.00 | 1.00 | 1.00 | 1.00 | 0.96 | 0.94 | 0.95 | 0.95 | 8.8 | 0.5 | 0.6 | 1.7 |
| also_see | 95 | 61 | 1.00 | 1.00 | 1.00 | 1.00 | 0.64 | 0.64 | 0.61 | 0.59 | 7.9 | 1.6 | 0.9 | 1.9 |
| instance_hypernym | 52 | 122 | 1.00 | 1.00 | 1.00 | 1.00 | 0.32 | 0.32 | 0.23 | 0.43 | 1.3 | 0.4 | 0.3 | 0.9 |
| member_of_domain_usage | 545 | 43 | 0.98 | 1.00 | 1.00 | 1.00 | 0.02 | 0.00 | 0.02 | 0.00 | 1.5 | 0.6 | 0.0 | 0.3 |
| member_of_domain_region | 543 | 42 | 0.88 | 0.89 | 1.00 | 0.93 | 0.02 | 0.02 | 0.00 | 0.02 | 1.0 | 0.4 | 0.8 | 0.7 |
| synset_domain_topic_of | 13 | 115 | 1.00 | 1.00 | 1.00 | 1.00 | 0.42 | 0.10 | 0.14 | 0.47 | 0.7 | 0.6 | 0.1 | 0.2 |
| member_meronym | 1402 | 307 | 1.00 | 1.00 | 1.00 | 1.00 | 0.22 | 0.02 | 0.01 | 0.22 | 7.9 | 3.4 | 1.5 | 5.6 |
| has_part | 848 | 196 | 1.00 | 1.00 | 1.00 | 1.00 | 0.24 | 0.05 | 0.09 | 0.22 | 7.1 | 2.4 | 1.3 | 3.9 |
| hypernym | 57 | 1254 | 1.00 | 1.00 | 1.00 | 1.00 | 0.15 | 0.02 | 0.02 | 0.22 | 3.7 | 1.2 | 0.0 | 1.7 |
| all | 5284 | 3306 | 0.99 | 0.99 | 1.00 | 0.99 | 0.47 | 0.37 | 0.37 | 0.50 | 5.9 | 1.2 | 0.5 | 2.1 |

**(a)** derivationally_related_form (R)   **(b)** instance_hypernym (S)   **(c)** synset_domain_topic_of (C)

**Figure 3:** Histograms of MuRE predictions for an example WN18RR relation of each type, split into true training, true test/validation and other instances.

models achieve almost perfect training accuracy. The additive-multiplicative MuRE gives best test set performance, followed (surprisingly) closely by MuRE$_I$, with multiplicative models (DistMult and TuckER) performing poorly on all but type R relations. Analysing a sample of "other" positive predictions for a relation of each type (see Appx. G), we estimate that TuckER is relatively accurate but pessimistic ($\sim 0.3$ correct of the 0.5 predictions $\approx 60\%$), MuRE$_I$ is optimistic but inaccurate ($\sim 2.3$ of $7.5 \approx 31\%$), whereas MuRE is both optimistic and accurate ($\sim 1.1$ of $1.5 \approx 73\%$).

Fig 3 shows histograms of MuRE prediction probabilities for the same sample relations, split by known truths (training and test/validation) and other instances. There is a clear distinction between relation types: for type R, most train and test triples are classified correctly with high confidence; for types S and C, an increasing majority of incorrect test predictions are far below the decision boundary, i.e. the model is confidently incorrect. For relation types where the model is less accurate, fewer positive predictions are made overall and the prediction distribution is more peaked towards zero.

This analysis probes further into the difficulty models have representing type S/C relations. Given the additional insight provided and the benefit of standalone predictions, we recommend the inclusion of predictive performance in future link prediction work.

## 5 CONCLUSION

Many models learn low-rank representations for knowledge graph link prediction, yet little is known about the latent structure they learn. We build on recent understanding of PMI-based word embeddings to theoretically establish what a relation representation must achieve to map a word embedding to those it is related to for the relations of knowledge graphs (relation conditions). Such conditions partition relations into three types and also provide a framework to assess loss functions of knowledge graph models. Any model that satisfies a relation's conditions can represent it if its entity embeddings are set to PMI-based word embeddings, i.e. a solution is known to exist. Whilst knowledge graph models do not learn the parameters of word embeddings, we show that the better a model's architecture satisfies a relation's conditions, the better its link prediction performance, fitting the premise that similar latent structure is exploited. Overall, we extend previous understanding of how semantic relations are encoded in relationships between PMI-based word embeddings – generalising from a limited set, e.g. similarity and analogy; we demonstrate commonality between the latent structure learned by PMI-based word embeddings (e.g. W2V) and knowledge graph representation models; and we provide novel insight into knowledge graph models by evaluating their predictive performance.

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

## A  CATEGORISING WORDNET RELATIONS

Table 7 describes how each WN18RR relation was assigned to its respective category.

**Table 7:** Explanation for the WN18RR relation category assignment.

| Type | Relation | Relatedness | Subject Specifics | Object Specifics |
|---|---|---|---|---|
| R | verb_group 
 derivationally_related_form 
 also_see | both verbs; similar in meaning 
 different syntactic categories; semantically related 
 semantically similar | - 
 - 
 - | - 
 - 
 - |
| S | hypernym 
 instance_hypernym | semantically similar 
 semantically similar | instance of the object 
 instance of the object | - 
 - |
| C | member_of_domain_usage 
 member_of_domain_region 
 member_meronym 
 has_part 
 synset_domain_topic_of | loosely semantically related (word usage features) 
 loosely semantically related (regional features) 
 semantically related 
 semantically related 
 semantically related | usage descriptor 
 region descriptor 
 collection of objects 
 collection of objects 
 broad feature set | broad feature set 
 broad feature set 
 part of the subject 
 part of the subject 
 domain descriptor |

## B  ANALYSING NELL-995 RELATIONS

**NELL-995:** We categorise a random subsample of 12 relations from the NELL-995 dataset (Xiong et al., 2017) containing 75,492 entities and 200 relations (a subset of NELL (Carlson et al., 2010)), which span our identified relation types (see Table 8). Explanation for the relation category assignment is shown in Table 9.

**Table 8:** Categorisation of NELL-995 relations.

| Type | Relation | Examples *(subject entity, object entity)* |
|---|---|---|
| R | teamplaysagainstteam 
 clothingtogowithclothing 
 agentcollaborateswithagent | *(sportsteam_rangers, sportsteam_mariners)*, *(sportsteam_phillies, sportsteam_tampa_bay_rays)* 
 *(clothing_shirts, clothing_trousers)*, *(clothing_shoes, clothing_black_shirt)* 
 *(musicartist_white_stripes, musicartist_jack_white)*, *(politician_obama, politician_hillary_clinton)* |
| S | professionistypeofprofession 
 animalistypeofanimal 
 chemicalistypeofchemical | *(profession_trial_lawyers, profession_attorneys)*, *(profession_engineers, profession_experts)* 
 *(mammal_cats, mammal_small_animals)*, *(bird_chickens, agriculturalproduct_livestock)* 
 *(chemical_moisture, chemical_gas)*, *(chemical_oxide, chemical_materials)* |
| C | athleteplayssport 
 itemfoundinroom 
 bodypartcontainsbodypart 
 atdate 
 locationlocatedwithinlocation 
 atlocation | *(athlete_joe_smith, sport_baseball)*, *(athlete_chris_cooley, sport_football)* 
 *(bedroomitem_bed, room_den)*, *(hallwayitem_refrigerator, visualizablething_kitchen_area)* 
 *(bodypart_system002, braintissue_eyes)*, *(bodypart_blood, bodypart_left_ventricle)* 
 *(currency_scotland, date_n2009)*, *(governmentorganization_wto, date_n2003)* 
 *(city_medellin, country_colombia)*, *(city_jackson, stateorprovince_wyoming)* 
 *(city_ogunquin, stateorprovince_maine)*, *(city_palmer_lake, stateorprovince_colorado)* |

**Table 9:** Explanation for the NELL-995 relation category assignment.

| Type | Relation | Relatedness | Subject Specifics | Object Specifics |
|---|---|---|---|---|
| R | teamplaysagainstteam 
 clothingtogowithclothing 
 agentcollaborateswithagent | both sport teams 
 both items of clothing that go together 
 both people or companies; related industries | - 
 - 
 - | - 
 - 
 - |
| S | professionistypeofprofession 
 animalistypeofanimal 
 chemicalistypeofchemical | semantically related (both profession types) 
 semantically related (both animals) 
 semantically related (both chemicals) | instance of the object 
 instance of the object 
 instance of the object | - 
 - 
 - |
| C | athleteplayssport 
 itemfoundinroom 
 bodypartcontainsbodypart 
 atdate 
 locationlocatedwithinlocation 
 atlocation | semantically related (sports features) 
 semantically related (room/furniture features) 
 emantically related (specific body part features) 
 loosely semantically related (start date features) 
 semantically related (geographical features) 
 semantically related (geographical features) | athlete descriptor 
 item descriptor 
 collection of objects 
 broad feature set 
 part of the subject 
 part of the subject | sport descriptor 
 room descriptor 
 part of the subject 
 date descriptor 
 collection of objects 
 collection of objects |

## C  SPLITTING THE NELL-995 DATASET

The test set of NELL-995 created by Xiong et al. (2017) contains only 10 out of 200 relations present in the training set. To ensure a fair representation of all training set relations in the validation and test sets, we create new validation and test set splits by combining the initial validation and test sets with the training set and randomly selecting 20,000 triples each from the combined dataset.

## D    IMPLEMENTATION DETAILS

All algorithms are re-implemented in PyTorch with the Adam optimizer (Kingma & Ba, 2015) that minimises binary cross-entropy loss, using hyper-parameters that work well for all models (learning rate: 0.001, batch size: 128, number of negative samples: 50). Entity and relation embedding dimensionality is set to $d_e = d_r = 200$ for all models except TuckER, for which $d_r = 30$ (Balažević et al., 2019b).

## E    KRACKHARDT HIERARCHY SCORE

The Krackhardt hierarchy score measures the proportion of node pairs $(x, y)$ where there exists a directed path $x \rightarrow y$, but not $y \rightarrow x$; and it takes a value of one for all directed acyclic graphs, and zero for cycles and cliques (Krackhardt, 2014; Balažević et al., 2019a).

Let $\boldsymbol{M} \in \mathbb{R}^{n \times n}$ be the binary *reachability matrix* of a directed graph $\mathcal{G}$ with $n$ nodes, with $\boldsymbol{M}_{i,j} = 1$ if there exists a directed path from node $i$ to node $j$ and 0 otherwise. The Krackhardt hierarchy score of $\mathcal{G}$ is defined as:

$$\text{Khs}_{\mathcal{G}} = \frac{\sum_{i=1}^{n} \sum_{j=1}^{n} \mathbb{1}(\boldsymbol{M}_{i,j} == 1 \wedge \boldsymbol{M}_{j,i} == 0)}{\sum_{i=1}^{n} \sum_{j=1}^{n} \mathbb{1}(\boldsymbol{M}_{i,j} == 1)}. \tag{1}$$

## F    SYMMETRY SCORE

The symmetry score $\in [-1, 1]$ (Hubert & Baker, 1979) for a relation matrix $\boldsymbol{R} \in \mathbb{R}^{d_e \times d_e}$ is defined as:

$$s = \frac{\sum \sum_{i \neq j} \boldsymbol{R}_{ij} \boldsymbol{R}_{ji} - \frac{(\sum \sum_{i \neq j} \boldsymbol{R}_{ij})^2}{d_e(d_e-1)}}{\sum \sum_{i \neq j} \boldsymbol{R}_{ij}^2 - \frac{(\sum \sum_{i \neq j} \boldsymbol{R}_{ij})^2}{d_e(d_e-1)}}, \tag{2}$$

where 1 indicates a symmetric and -1 an anti-symmetric matrix.

## G    "OTHER" PREDICTED FACTS

Tables 10 to 13 shows a sample of the unknown triples (i.e. those formed using the WN18RR entities and relations, but not present in the dataset) for the *derivationally_related form* (R), *instance_hypernym* (S) and *synset_domain_topic_of* (C) relations at a range of probabilities $(\sigma(\phi(e_s, r, e_o)) \approx \{0.4, 0.6, 0.8, 1\})$, as predicted by each model. True triples are indicated in bold; instances where a model predicts an entity is related to itself are indicated in blue.

Table 10: "Other" facts as predicted by MuRE_I.

| Relation (Type) | $\sigma(\phi(e_s,r,e_o)) \approx 0.4$ | $\sigma(\phi(e_s,r,e_o)) \approx 0.6$ | $\sigma(\phi(e_s,r,e_o)) \approx 0.8$ | $\sigma(\phi(e_s,r,e_o)) \approx 1$ |
|---|---|---|---|---|
| derivationally_related_form (R) | (equalizer_NN_2, set_off_VB_5)
(constellation_NN_2, satellite_NN_3)
(**shrink_VB_3, subtraction_NN_2**)
(continue_VB_10, proceed_VB_1)
(support_VB_6, defend_VB_5)
(shutter_NN_1, fill_up_VB_3)
(yawning_NN_1, patellar_reflex_NN_1)
(**yaw_NN_1, spiral_VB_1**)
(stratum_NN_2, social_group_NN_1)
(duel_VB_1, scuffle_NN_3) | (extrapolation_NN_1, maths_NN_1)
(spread_VB_5, circularize_VB_3)
(flaunt_NN_1, showing_NN_2)
(**extrapolate_VB_3, synthesis_NN_3**)
(strategist_NN_1, machination_NN_1)
(crush_VB_4, grind_VB_2)
(spike_VB_5, steady_VB_2)
(licking_NN_1, vanquish_VB_1)
(**synthetical_JJ_1, synthesizer_NN_2**)
(realization_NN_2, embodiment_NN_3) | (sewer_NN_2, stitcher_NN_1)
(lard_VB_1, vegetable_oil_NN_1)
(**snuggle_NN_1, draw_close_VB_3**)
(**train_VB_3, training_NN_1**)
(**scratch_VB_3, skin_sensation_NN_1**)
(scheme_NN_5, schematization_NN_1)
(ordain_VB_3, vest_VB_1)
(lie_VB_1, front_end_NN_1)
(tread_NN_1, step_NN_9)
(**register_NN_3, file_away_VB_1**) | (trail_VB_2, trail_VB_2)
(worship_VB_1, worship_VB_1)
(steer_VB_1, steer_VB_1)
(sort_out_VB_1, sort_out_VB_1)
(make_full_VB_1, make_full_VB_1)
(utilize_VB_1, utilize_VB_1)
(geology_NN_1, geology_NN_1)
(zoology_NN_2, zoology_NN_2)
(uranology_NN_1, uranology_NN_1)
(travel_VB_1, travel_VB_1) |
| instance_hypernym (S) | (thomas_aquinas_NN_1, martyr_NN_2)
(volcano_islands_NN_1, volcano_NN_2)
(cape_horn_NN_1, urban_center_NN_1)
(bergen_NN_1, national_capital_NN_1)
(marshall_NN_2, generalship_NN_1)
(**nansen_NN_1, venturer_NN_2**)
(wisconsin_NN_2, state_capital_NN_1)
(prussia_NN_1, stockade_NN_2)
(**de_mille_NN_1, dancing-master_NN_1**)
(aegean_sea_NN_1, aegean_island_NN_1) | (**taiwan_NN_1, asian_nation_NN_1**)
(**st._gregory_of_n._NN_1, canonization_NN_1**)
(st._gregory_of_n._NN_1, saint_VB_2)
(mccormick_NN_1, find_VB_8)
(**st._gregory_i_NN_1, bishop_NN_1**)
(richard_buckminster_f._NN_1, technological_JJ_2)
(thomas_aquinas_NN_1, archbishop_NN_1)
(**marshall_NN_2, general_officer_NN_1**)
(newman_NN_2, primateship_NN_1)
(thomas_the_apostle_NN_1, sanctify_VB_1) | (**prophets_NN_1, gospels_NN_1**)
(malcolm_x_NN_1, passive_resister_NN_1)
(taiwan_NN_1, national_capital_NN_1)
(truth_NN_5, abolitionism_NN_1)
(**thomas_aquinas_NN_1, saint_VB_2**)
(central_america_NN_1, s._am._nation_NN_1)
(de_mille_NN_1, dance_VB_1)
(st._gregory_i_NN_1, apostle_NN_3)
(fertile_crescent_NN_1, asian_nation_NN_1)
(robert_owen_NN_1, industry_NN_1) | (**helsinki_NN_1, urban_center_NN_1**)
(mannheim_NN_1, stockade_NN_2)
(**nippon_NN_1, nippon_NN_1**)
(victor_hugo_NN_1, novel_NN_1)
(regiomontanus_NN_1, uranology_NN_1)
(**prophets_NN_1, book_NN_10**)
(thomas_aquinas_NN_1, church_father_NN_1)
(woody_guthrie_NN_1, minstrel_VB_1)
(central_america_NN_1, c._am._nation_NN_1)
(aegean_sea_NN_1, island_NN_1) |
| synset_domain_topic_of (C) | (write_VB_8, tape_VB_3)
(introvert_NN_1, scientific_discipline_NN_1)
(**libel_NN_1, slur_NN_2**)
(etymologizing_NN_1, law_NN_1)
(**temple_NN_4, place_of_worship_NN_1**)
(trial_impression_NN_1, proof_VB_1)
(friend_of_the_court_NN_1, war_machine_NN_1)
(**multiv._analysis_NN_1, applied_math_NN_1**)
(**sell_VB_1, transaction_NN_1**)
(draw_VB_6, represent_VB_9) | (draw_VB_6, creative_person_NN_1)
(**suborder_NN_1, taxonomic_group_NN_1**)
(draw_VB_6, draw_VB_6)
(**first_sacker_NN_1, ballgame_NN_2**)
(alchemize_VB_1, modify_VB_3)
(sermon_NN_1, sermon_NN_1)
(**saint_VB_2, catholic_church_NN_1**)
(male_JJ_1, masculine_JJ_2)
(fire_VB_3, zoology_NN_2)
(sell_VB_1, sell_VB_1) | (libel_NN_1, sully_VB_3)
(relationship_NN_4, relationship_NN_4)
(**etymologizing_NN_1, linguistics_NN_1**)
(turn_VB_12, cultivation_NN_2)
(brynhild_NN_1, mythologize_VB_2)
(**brynhild_NN_1, myth_NN_1**)
(**assist_NN_2, am._football_game_NN_1**)
(mitzvah_NN_2, human_activity_NN_1)
(drive_NN_12, drive_VB_8)
(**relationship_NN_4, biology_NN_1**) | (**libel_NN_1, disparagement_NN_1**)
(**roll-on_roll-off_NN_1, transport_NN_1**)
(**prance_VB_4, equestrian_sport_NN_1**)
(**libel_NN_1, traducement_NN_1**)
(**sell_VB_1, selling_NN_1**)
(trot_VB_2, ride_horseback_VB_1)
(prance_VB_4, ride_horseback_VB_1)
(gallop_VB_1, ride_horseback_VB_1)
(**brynhild_NN_1, mythology_NN_2**)
(**drive_NN_12, badminton_NN_1**) |

Table 11: "Other" facts as predicted by DistMult.

| Relation (Type) | $\sigma(\phi(e_s, r, e_o)) \approx 0.4$ | $\sigma(\phi(e_s, r, e_o)) \approx 0.6$ | $\sigma(\phi(e_s, r, e_o)) \approx 0.8$ | $\sigma(\phi(e_s, r, e_o)) \approx 1$ |
|---|---|---|---|---|
| derivationally_related_form (R) | (stag_VB_3, undercover_work_NN_1)
(print_VB_4, publisher_NN_2)
(crier_NN_3, pitchman_NN_2)
(play_VB_26, turn_NN_10)
(count_VB_4, recite_VB_2)
(vividness_NN_2, imbue_VB_3)
(sea_maw_NN_1, larus_NN_1)
(alkali_NN_2, acidify_VB_2)
(see_VB_17, understand_VB_2)
(shun_VB_1, hedging_NN_2) | (dish_NN_2, stew_NN_2)
(expose_VB_3, show_NN_1)
(system_NN_9, orderliness_NN_1)
(spread_NN_4, strew_VB_1)
(take_down_VB_2, put_VB_2)
(wrestle_VB_4, wrestler_NN_1)
(autotr._organism_NN_1, epiphytic_JJ_1)
(duel_VB_1, slugfest_NN_1)
(vocal_NN_2, rock_star_NN_1)
(smelling_NN_1, scent_VB_1) | (shrink_NN_1, pedology_NN_1)
(finish_VB_6, finishing_NN_2)
(play_VB_26, playing_NN_3)
(centralization_NN_1, unite_VB_6)
(existence_NN_1, living_NN_3)
(mouth_VB_3, sassing_NN_1)
(constellation_NN_2, star_NN_1)
(print_VB_4, publishing_house_NN_1)
(puzzle_VB_2, secret_NN_3)
(uranology_NN_1, tt_NN_1) | (alliterate_VB_1, versifier_NN_1)
(geology_NN_1, structural_JJ_5)
(resect_VB_1, amputation_NN_2)
(nutrition_NN_3, man_NN_4)
(saint_NN_3, sanctify_VB_1)
(right_fielder_NN_1, leftfield_NN_1)
(list_VB_4, slope_NN_2)
(lieutenancy_NN_1, captain_NN_1)
(tread_NN_1, step_VB_7)
(exenteration_NN_1, enucleate_VB_2) |
| instance_hypernym (S) | (wisconsin_NN_2, urban_center_NN_1)
(marshall_NN_2, lieutenant_general_NN_1)
(abidjan_NN_1, cote_d'ivoire_NN_1)
(world_war_i_NN_1, urban_center_NN_1)
(st._paul_NN_2, evangelist_NN_2)
(deep_south_NN_1, urban_center_NN_1)
(nuptse_NN_1, urban_center_NN_1)
(ticino_NN_1, urban_center_NN_1)
(aegean_sea_NN_1, aegean_island_NN_1)
(cowpens_NN_1, war_of_am._ind._NN_1) | (mississippi_river_NN_1, american_state_NN_1)
(r._e._byrd_NN_1, commissioned_officer_NN_1)
(kobenhavn_NN_1, urban_center_NN_1)
(the_gambia_NN_1, africa_NN_1)
(tirich_mir_NN_1, urban_center_NN_1)
(r._e._byrd_NN_1, military_advisor_NN_1)
(r._e._byrd_NN_1, aide-de-camp_NN_1)
(tampa_bay_NN_1, urban_center_NN_1)
(tidewater_region_NN_1, south_NN_1)
(r._e._byrd_NN_1, executive_officer_NN_1) | (deep_south_NN_1, south_NN_1)
(capital_of_gambia_NN_1, urban_center_NN_1)
(south_west_africa_NN_1, africa_NN_1)
(brandenburg_NN_1, urban_center_NN_1)
(sierra_nevada_NN_1, urban_center_NN_1)
(malcolm_x_NN_1, emancipationist_NN_1)
(north_plate_river_NN_1, urban_center_NN_1)
(oslo_NN_1, urban_center_NN_1)
(zaire_river_NN_1, urban_center_NN_1)
(transylvanian_alps_NN_1, urban_center_NN_1) | (helsinki_NN_1, urban_center_NN_1)
(the_nazarene_NN_1, save_VB_7)
(irish_capital_NN_1, urban_center_NN_1)
(r._e._byrd_NN_1, inspector_general_NN_1)
(r._e._byrd_NN_1, chief_of_staff_NN_1)
(central_america_NN_1, c._am._nation_NN_1)
(malcolm_x_NN_1, environmentalist_NN_1)
(the_nazarene_NN_1, christian_JJ_1)
(thomas_aquinas_NN_1, church_father_NN_1)
(the_nazarene_NN_1, el_nino_NN_2) |
| synset_domain_topic_of (C) | (limitation_NN_4, trammel_VB_2)
(light_colonel_NN_1, colonel_NN_1)
(nurse_VB_1, nursing_NN_1)
(sermon_NN_1, prophesy_VB_2)
(libel_NN_1, practice_of_law_NN_1)
(slugger_NN_1, baseball_player_NN_1)
(rma_NN_1, chemistry_NN_1)
(metrify_VB_1, versify_NN_1)
(trial_impression_NN_1, publish_VB_1)
(turn_VB_12, plowman_NN_1) | (roll-on_roll-off_NN_1, transport_NN_1)
(hizb_ut-tahrir_NN_1, asia_NN_1)
(slugger_NN_1, softball_game_NN_1)
(sermon_NN_1, sermonize_VB_1)
(draw_VB_6, drawer_NN_3)
(turn_VB_12, plow_NN_1)
(assist_NN_2, softball_game_NN_4)
(council_NN_2, assembly_NN_1)
(throughput_NN_1, turnout_NN_4)
(cream_VB_1, cream_NN_2) | (etymologizing_NN_1, explanation_NN_1)
(ferry_VB_3, travel_VB_1)
(public_prosecutor_NN_1, prosecute_VB_2)
(alchemize_VB_1, modify_VB_3)
(libel_NN_1, libel_VB_1)
(turn_VB_12, till_VB_1)
(hit_NN_1, hit_VB_1)
(fire_VB_3, flaming_NN_1)
(ring_NN_4, chemical_chain_NN_1)
(libidinal_energy_NN_1, charge_NN_9) | (flat_JJ_5, matte_NN_2)
(etymologizing_NN_1, derive_VB_3)
(hole_out_VB_1, hole_NN_3)
(relationship_NN_4, relation_NN_1)
(drive_NN_12, badminton_NN_1)
(etymologizing_NN_1, etymologize_VB_2)
(matrix_algebra_NN_1, diagonalization_NN_1)
(cabinetwork_NN_2, woodworking_NN_1)
(cabinetwork_NN_2, bottom_VB_1)
(cabinetwork_NN_2, upholster_VB_1) |

**Table 12:** "Other" facts as predicted by TuckER.

| Relation (Type) | $\sigma(\phi(e_s, r, e_o)) \approx 0.4$ | $\sigma(\phi(e_s, r, e_o)) \approx 0.6$ | $\sigma(\phi(e_s, r, e_o)) \approx 0.8$ | $\sigma(\phi(e_s, r, e_o)) \approx 1$ |
|---|---|---|---|---|
| derivationally_related_form (R) | (tympanist_NN_1, gong_NN_2)
(indication_NN_1, signalize_VB_2)
(turn_over_VB_3, rotation_NN_3)
(date_VB_5, geological_dating_NN_1)
(set_VB_23, emblem_NN_2)
(tyro_NN_1, start_VB_5)
(identification_NN_1, name_VB_5)
(stabber_NN_1, thrust_VB_5)
(justification_NN_1, apology_NN_2)
(manufacture_VB_1, prevarication_NN_1) | (mash_NN_2, mill_VB_2)
(walk_VB_9, zimmer_frame_NN_1)
(use_VB_5, utility_NN_2)
(musical_instrument_NN_1, write_VB_6)
(lining_NN_3, wrap_up_VB_1)
(scrap_VB_2, struggle_NN_2)
(tape_VB_3, tape_recorder_NN_1)
(vindicate_VB_2, justification_NN_2)
(teaching_NN_1, percolate_VB_3)
(synchronize_VB_2, synchroscope_NN_1) | (take_chances_VB_1, venture_NN_1)
(shutter_NN_1, fill_up_VB_3)
(exit_NN_3, leave_VB_1)
(trembler_NN_1, vibrate_VB_1)
(motivator_NN_1, trip_VB_4)
(support_VB_6, indorsement_NN_1)
(federate_VB_2, confederation_NN_1)
(take_over_VB_6, return_NN_7)
(wait_on_VB_1, supporter_NN_3)
(denote_VB_3, promulgation_NN_1) | (venturer_NN_2, venturer_NN_2)
(dynamitist_NN_1, dynamitist_NN_1)
(love_VB_3, lover_NN_2)
(snuggle_NN_1, squeeze_VB_8)
(departed_NN_1, die_VB_2)
(position_VB_1, placement_NN_1)
(repentant_JJ_1, repentant_JJ_1)
(tread_NN_1, step_VB_7)
(stockist_NN_1, stockist_NN_1)
(philanthropist_NN_1, philanthropist_NN_1) |
| instance_hypernym (S) | (deep_south_NN_1, south_NN_1)
(st_paul_NN_2, organist_NN_1)
(helsinki_NN_1, urban_center_NN_1)
(malcolm_x_NN_1, emancipationist_NN_1)
(thomas_the_apostle_NN_1, church_father_NN_1)
(st_gregory_of_n._NN_1, sermonizer_NN_1)
(robert_owen_NN_1, movie_maker_NN_1)
(theresa_NN_1, monk_NN_1)
(st_paul_NN_2, philosopher_NN_1)
(ibn-roshd_NN_1, pedagogue_NN_1) | (thomas_aquinas_NN_1, bishop_NN_1)
(irish_capital_NN_1, urban_center_NN_1)
(thomas_the_apostle_NN_1, apostle_NN_2)
(st_paul_NN_2, apostle_NN_3)
(mccormick_NN_1, painter_NN_1)
(thomas_the_apostle_NN_1, troglodyte_NN_1)
(mccormick_NN_1, electrical_engineer_NN_1)
(mississippi_river_NN_1, american_state_NN_1) | (cowpens_NN_1, siege_NN_1)
(mccormick_NN_1, arms_manufacturer_NN_1)
(thomas_the_apostle_NN_1, evangelist_NN_2)
(mccormick_NN_1, technologist_NN_1)
(st_gregory_i_NN_1, church_father_NN_1) | (t._e._byrd_NN_1, siege_NN_1)
(shaw_NN_3, women's_rightist_NN_1)
(aegean_sea_NN_1, aegean_island_NN_1)
(thomas_aquinas_NN_1, church_father_NN_1) |
| synset_domain_topic_of (C) | (roll-on_roll-off_NN_1, motorcar_NN_1)
(libel_NN_1, legislature_NN_1)
(roll-on_roll-off_NN_1, passenger_vehicle_NN_1) | (drive_NN_12, badminton_NN_1) | | |

Table 13: "Other" facts as predicted by MuRE.

| Relation (Type) | $\sigma(\phi(e_s, r, e_o)) \approx 0.4$ | $\sigma(\phi(e_s, r, e_o)) \approx 0.6$ | $\sigma(\phi(e_s, r, e_o)) \approx 0.8$ | $\sigma(\phi(e_s, r, e_o)) \approx 1$ |
|---|---|---|---|---|
| derivationally_related_form (R) | (surround_VB_1, wall_NN_1)
(unpleasant_JJ_1, unpalatableness_NN_1)
(love_VB_3, enjoyment_NN_2)
(magnitude_NN_1, tall_JJ_1)
(testify_VB_2, information_NN_1)
(connect_VB_6, converging_NN_1)
(connect_VB_6, connexion_NN_4)
(operate_VB_4, psyop_NN_1)
(market_VB_1, trade_NN_4)
(operate_VB_4, mission_NN_2) | (word_picture_NN_1, sketch_VB_2)
(develop_VB_10, adjustment_NN_4)
(gloss_VB_3, commentary_NN_1)
(violate_VB_2, violation_NN_3)
(suffocate_VB_1, strangler_tree_NN_1)
(number_VB_3, point_NN_12)
(develop_VB_10, organic_process_NN_1)
(plication_NN_1, twist_VB_4)
(split_up_VB_3, separation_NN_5)
(plication_NN_1, wrinkle_VB_2) | (smelling_NN_1, wind_VB_4)
(try_out_VB_1, somatic_cell_nuclear_transplantation_NN_1)
(lighting_NN_4, set_on_fire_VB_1)
(symphalangus_NN_1, one-half_NN_1)
(just_JJ_3, validity_NN_1)
(reprove_VB_1, talking_to_NN_1)
(sustain_VB_5, beam_NN_2)
(spring_NN_6, hurdle_VB_1)
(spark_NN_1, scintillae_VB_1)
(utility_NN_2, functional_JJ_1) | (spoliation_NN_2, sack_VB_1)
(desire_NN_2, hope_VB_2)
(snuffle_VB_3, whine_NN_1)
(nasalization_NN_1, sound_out_VB_1)
(tread_NN_1, step_VB_7)
(yearn_VB_1, pining_NN_1)
(unreliableness_NN_1, arbitrary_JJ_1)
(travesty_NN_2, travesty_NN_2)
(spark_NN_1, sparkle_VB_1)
(stockist_NN_1, stockist_NN_1) |
| instance_hypernym (S) | (malcolm_x_NN_1, hipster_NN_1)
(the_nazarene_NN_1, judaism_NN_2)
(old_line_state_NN_1, river_NN_1)
(r._e._byrd_NN_1, commissioned_officer_NN_1)
(south_korea_NN_1, peninsula_NN_1)
(st._gregory_of_n._NN_1, vicar_of_christ_NN_1)
(nippon_NN_1, italian_region_NN_1)
(robert_owen_NN_1, tycoon_NN_1)
(mandalay_NN_1, national_capital_NN_1)
(nan_ling_NN_1, urban_center_NN_1) | (central_america_NN_1, central_america_NN_1)
(st._gregory_i_NN_1, church_father_NN_1)
(south_korea_NN_1, african_nation_NN_1)
(malcolm_x_NN_1, passive_resister_NN_1)
(malcolm_x_NN_1, birth-control_reformer_NN_1)
(los_angeles_NN_1, port_NN_1)
(great_lakes_NN_1, canadian_province_NN_1)
(transylvanian_alps_NN_1, urban_center_NN_1)
(gettysburg_NN_2, siege_NN_1)
(wisconsin_NN_2, geographical_region_NN_1) | (theresa_NN_1, monk_NN_1)
(nippon_NN_1, european_nation_NN_1)
(great_lakes_NN_1, river_NN_1)
(r._e._byrd_NN_1, noncommissioned_officer_NN_1)
(world_war_i_NN_1, pitched_battle_NN_1)
(irish_capital_NN_1, urban_center_NN_1)
(volcano_islands_NN_1, urban_center_NN_1)
(nippon_NN_1, american_state_NN_1)
(helsinki_NN_1, urban_center_NN_1)
(capital_of_gambia_NN_1, urban_center_NN_1) | |
| synset_domain_topic_of (C) | (libel_NN_1, criminal_law_NN_1)
(brynhild_NN_1, mythology_NN_2)
(slugger_NN_1, sport_NN_1)
(sell_VB_1, law_NN_1)
(semitic_deity_NN_1, mythology_NN_1)
(nuclear_deterrence_NN_1, law_NN_1)
(reception_NN_5, baseball_game_NN_1)
(photosynthesis_NN_1, chemistry_NN_1)
(isolde_NN_1, parable_NN_1)
(assist_NN_2, court_game_NN_1) | (write_VB_8, transcription_NN_5)
(temple_NN_4, muslimism_NN_2)
(assist_NN_2, hockey_NN_1)
(relationship_NN_4, biology_NN_1)
(apostle_NN_3, western_church_NN_1)
(assist_NN_2, sport_NN_1)
(trot_VB_2, equestrian_sport_NN_1)
(rna_NN_1, chemistry_NN_1)
(assist_NN_2, soccer_NN_1)
(assist_NN_2, football_game_NN_1) | (assist_NN_2, am._football_game_NN_1)
(drive_NN_12, court_game_NN_1)
(sell_VB_1, offense_NN_3)
(slugger_NN_1, softball_game_NN_1)
(drive_NN_12, badminton_NN_1) | |

