# OpenReview forum: "On Understanding Knowledge Graph Representation"
_ICLR.cc/2020/Conference — Reject_

### Official Review · AnonReviewer3 · 2019-10-16
**Official Blind Review #3**

**Rating:** 6

**Review:**

Summary:
The paper attempts to understand the latent structure underlying knowledge graph embedding methods. The work can be seen as an extension of understanding of PMI-based word embedding methods. They categorize knowledge graph relations into three categories based on their relation conditions: Relatedness (R), Specialisation (S), and Context-shift (C). For each category, they evaluate a representative of different types of knowledge graph embedding methods. Through results, they demonstrate that a model’s ability to represent a specific relation type depends on the limitations imposed by the model architecture with respect to satisfying the necessary relation conditions.

Questions:
1. The results in Tables 3 and 4 demonstrate that MuRE is the most effective method for handling different types of relations but then how come its performance on FB15k-237 (.336 MRR) is significantly lower than other methods like TuckER (.358 MRR). Can you provide an explanation?

2. In Section 3.2, the authors list 4 predictions (P1-4). It would be great if authors could provide some more reasoning behind coming with these predictions.

3. In Section 4.2, it is stated that “ranking based metrics like MRR and hits@k are flawed if entities are related to more than k others”. It would be great if the authors could give an example to make it more clear.


**Experience Assessment:**

I have published one or two papers in this area.

**Review Assessment: Checking Correctness Of Derivations And Theory:**

I assessed the sensibility of the derivations and theory.

**Review Assessment: Checking Correctness Of Experiments:**

I assessed the sensibility of the experiments.

**Review Assessment: Thoroughness In Paper Reading:**

I read the paper at least twice and used my best judgement in assessing the paper.

---

> ### Author Response · Authors · 2019-11-08
> **Author response to Reviewer #3**
>
> Thank you for your review and the time taken for it, below we address each of your points in turn.
>
> 1) The difference in performance on the FB15k-237 (FB) dataset is due to multi-task learning (as discussed in the TuckER paper, Balaevic et al. (2019b)). Amongst the models considered, TuckER is the only one to share parameters between relations, which is encouraged within its core tensor due to its low-rank. As such, on datasets with many relations and relatively few instances per relation, e.g. FB and NELL, the multi-task learning ability of TuckER offers material advantage. We mention this (briefly) in respect of the the NELL dataset in section 4.1, which we have italicised to make more clear. For avoidance of doubt, this is not our reason for omitting results for the FB dataset, which is because the vast majority of FB relations are found to be of type C, whereas we analyse performance differences between relation types and hence use WN and NELL, which have a broader variety of relations by type.
>
> 2) we agree that rationale for the predictions could be more clear and have improved clarity (e.g. the start of Sec 3.2 and key contributions in Sec 1), to present the predictions is a clearer light. To expand on their rationale:
> P1: Type R is a special case of type S, which is a special case of type C (Section 3.1). Hence type S, for example, subsumes type R and (all else being equal) a type S relation requires more parameters to be learned than one of type R and, in that sense, is ``harder to learn’’. Further to this, the “shape” of the relation-mapping changes between relation types: type R require only a multiplicative (matrix) component, type S an extra additive (vector) component and type C a further additive component. Whether the architecture of a model supports a relation’s “shape” (or, in the language of P1, meet the “relation conditions) is expected to affect performance.
> P2: Since type R relations are (by definition) symmetric, their relation matrices must also be.
> P3: Since no additive components (“offset vectors”) are required for relatedness relations (type R), any vector norms for those relations are predicted to be small.
> P4: The strength (s) of the relatedness of a relation r is defined as |S| where S is the set of context words that both entities must co-occur with similarly for r to hold. In the full rank space of PMI vectors each word corresponds to a dimension, thus s is also the dimensionality of a common PMI vector component (i.e. the component in the dimensions of S). That common component can be tested for with a projection matrix of rank s. In the lower dimension of embeddings, the dimensionality of S is obscured, but is anticipated to be reflected in the eigenvalues of the relation matrix due to its relationship with the projection matrix (Section 3.2).
>
> 3) As an example of how hits@k metrics can be flawed: consider computing hits@3 for a model with a dataset whose entities include {UK, London, Edinburgh, Brighton, Manchester, York, Birmingham}, training set contains (UK, contains_city, Edinburgh), (UK, contains_city, Manchester) and test set contains (UK, contains_city, London).
>
> Each test triple is evaluated by removing an entity and ranking the score assigned to that “true” entity amongst all entities in the dataset, excluding other known true triples (i.e. “filtering”). When the test triple (UK, contains_city, London) is evaluated by removing London, let the top scoring entities be, in descending order:
>   Edinburgh, Brighton, Manchester, York, London, Birmingham, ...
> After removing Edinburgh and Manchester (as known “true”s), the order is:
>   Brighton, York, London, Birmingham, ...
> And the sought answer (London) appears in the top k (i.e. 3), and contributes to hits@k metric.
> However, if (UK, contains_city, Edinburgh) happened to not be in the dataset, the order would be:
>   Edinburgh, Brighton, York, London, Birmingham, ...
> And the sought answer would not appear in the top k, even though all top 5 answers are correct.
>
> This demonstrates that ranking metrics for 1-to-many, many-to-many or many-to-1 relations can be affected (arbitrarily) by unknown true answers, which cannot be tested for or evaluated (without further annotation), since they are by definition unknown. Further, such instances are always assumed to exist since the central aim of link prediction is to predict them, i.e. true facts that are not previously known.

---

### Official Review · AnonReviewer2 · 2019-10-22
**Official Blind Review #2**

**Rating:** 6

**Review:**

This paper proposes to provide a detailed study on the explainability of link prediction (LP) models by utilizing a recent interpretation of word embeddings. More specifically, the authors categorize the relations in KG into three categories (R, S, C) using the correlation between the semantic relation between two words and the geometric relationship between their embeddings. The authors utilize this categorization to provide a better understanding of LP models’ performance through several experiments.

This paper reads well and the results appear sound. I personally believe that works on better understanding KGC models are a very essential direction which is mostly ignored in this field of study. Moreover, the provided experiments support the authors’ intuition and arguments.

As for the drawbacks, I find the technical novelty of the paper is somewhat limited, as the proposed method consists of a mostly straightforward combination of existing methods. Further, I believe this work needs more experimental results and decisive conclusions identifying future directions to achieve better performance on link prediction. My concerns are as follows:

•    I am wondering about the reason for omitting Max/Avg path for two of the relations in WN18RR? Further, the average of 15.2 for the shortest path between entities with “also_see” relation appears to be a mistake?
•    Was there any specific reason in choosing WN18RR and NELL-995 KGs for the experiments?
•    It would be interesting to see the length of paths between entities for train and test data separately.
•    I suggest providing a statistical significance evaluation for each experiment to better validate the conclusions.
•    I find the provided study in section 4.2 very similar to the triple classification task in KGs. Can you elaborate on the differences and potential advantages of your setting?
•    I am wondering how you identified the “Other True” triples for WN18RR KG in section 4.2 experiments?

On overall, although I find the proposed study very interesting and enlightening, I believe that the paper needs more experimental results and decisive conclusions.


**Experience Assessment:**

I have published one or two papers in this area.

**Review Assessment: Checking Correctness Of Derivations And Theory:**

I assessed the sensibility of the derivations and theory.

**Review Assessment: Checking Correctness Of Experiments:**

I assessed the sensibility of the experiments.

**Review Assessment: Thoroughness In Paper Reading:**

I read the paper thoroughly.

---

> ### Author Response · Authors · 2019-11-08
> **Author response to Reviewer #2: conclusion**
>
> Re “decisive conclusions”: we agree that the conclusions of the paper are insufficiently clear and have improved the paper to address. Decisive conclusions that we make:
>   1) previous understanding of how semantic relations are encoded between PMI-based word embeddings for a few relations (e.g. similarity, analogies, etc - Allen & Hospedale (2019), Allen et al. (2019)) is extended to derive the difference between word embeddings for the general relations of knowledge graphs, which translate into linear algebraic mappings. From their mappings, relations can be categorised into 3 types and components of the mappings (e.g. projection matrix, translation vector) relate to meaningful/interpretable semantic aspects of the relation (e.g. relatedness between entities, entity-specific features).
>   2) that PMI-based word embeddings and knowledge graph entity embeddings show commonality to their latent structure — despite the significant differences between their training data and methodology. We demonstrate this by: (i) deriving properties of the relation mappings  (based on word embeddings), e.g. vector norm, matrix symmetry/effective rank, and identifying those in actual knowledge graph representations; and (ii) showing that the relative performance of knowledge graph models for each relation type accords with how well a model’s architecture satisfies the corresponding relation conditions (based on word embeddings).
>   3) that stand-alone classification performance should be evaluated for future models since the task itself may be of more practical use than ranking metrics, and it provides novel insight into model performance.
>   Overall, we provide an important step towards a theoretical understanding of the latent structure of knowledge graph representations. In terms of practical use, our results: provide understanding as to which model is most appropriate for a new dataset (e.g. if relations were known, a priori, to be symmetric); suggest that different aspects of relations (e.g. type, strength of relatedness) could be quantitatively evaluated; and indicate where future research effort might be directed (e.g. type C relations). Furthermore, whilst whether multiplicative (e.g. DistMult) or additive (e.g. TransE) link prediction models are superior has been an open question, we now provide theoretical justification that the answer is both (e.g. as in MuRE).

---

> ### Author Response · Authors · 2019-11-08
> **Author response to Reviewer #2: concerns**
>
> Thank you for your review and the time taken for it, below we address each of your points in turn.
>
> * Re “technical novelty of … proposed method”: we are delighted that our research direction is considered essential and well regarded. To be clear, we do not propose a new method here, rather we derive, using word embeddings, theoretical conditions that a relation representation must satisfy for the relations of knowledge graphs. Amongst other things, we then demonstrate that the performance of recent knowledge graph models corresponds with their ability to satisfy those conditions, demonstrating a commonality between the interpretable latent space of PMI-based word embeddings and the previously “blackbox” latent space of knowledge graphs. We agree that our key findings were insufficiently clear and have improved clarity (as below).
>
> * Re “omitting Max/Avg path”: by definition (see also the MuRE paper, Balazevic et al. (2019a)), where a Kh score is zero the relation is not tree-structured and has no extended paths. We have changed this to “1” for greater clarity.
>
> * Re “also_see” path lengths: we agree this is a notable outlier, but correctly reflects the data (see also the MuRE paper) indicating that the relation subgraph contains long chains.
>
> * Re “choosing WN18RR and NELL-995 KGs”: these datasets contain a relatively broad spread of relations by type and so provide the best testbed to demonstrate differences in model performance across relation types. By comparison, the FB15k-237 dataset (FB) contains almost all type C relations. We agree that is not sufficiently clear and have moved the explanation from Appx B to Sec 3.1.
>
> * Re “significance of conclusions”: we agree with the importance and confirm that, due to low stochasticity of the algorithms, variance between model runs is broadly negligible and error bars are typically omitted in the literature. One recent work “Hypernetwork Knowledge Graph Embeddings” (Balazevic et al. (2019)) that does, reports standard deviations across 5 model runs of at most 0.003 for the metrics we consider (their Tables 7 & 8), which is consistent with our experience.
> Results aggregated by model and relation type that more succinctly validate our conclusions are included in our response to Reviewer #1.
>
> * Re “triple classification task”: Standard metrics in the link prediction literature are MRR and Hits@k based on the ranking of the score attributed to the sought answer amongst the scores of all possible answers. A typical classification task would assign a stand-alone prediction (in [0,1]), which, to the best of our knowledge, rarely appears in the link prediction literature. Indeed, since datasets (e.g. WN, FB) typically contain only positive samples (true facts), it is not straightforward to evaluate model classification performance since perfect test set accuracy is achieved trivially by predicting all relations to be true (i.e. a high false positive rate). We address this by assessing the truth of a random sample of the triples each model predicts as true but for which the model has no ground truth (“other true”) (Sec 4.2).
>
> * Re “other true”: for a given (subject_entity, relation) test pair, a model predicts the truth for each object_entity of the knowledge base ( an evaluation triple). Any evaluation triple that appears in the training or test set, and so known to be true, contributes to “Accuracy (train)” or “Accuracy (test)”, resp. The remaining majority of triples are unknown to be true/false (although typically the vast majority are false) - any a model predicts to be true we term “other true”. As above, we review a random sample of these for each model to estimate model precision. Note that predicting such “truly unknown” instances is the ultimate aim of link prediction so should not be overlooked.

---

### Official Review · AnonReviewer1 · 2019-10-24
**Official Blind Review #1**

**Rating:** 6

**Review:**


There has been a large family of knowledge base models developed in the recent years, aiming to encode both entities and relations in a latent space, so the entities can be “linked” via a relation-specific mapping.

The paper focuses on understanding these entity embeddings (and geometric embedding relationships), built on top of the connections with PMI-based word embeddings.

The paper categorizes all the relations into 3 types (1) related, (2) specialisation, and (3) context shift, and examines some relations in WordNet and NELL, and then empirically evaluates the performance of different types of models and draws the correlation of the results and intuitive understanding of different types of relations.

To me, this paper is more like providing some intuitive explanations of existing KG embeddings methods and their performance (not really theoretical justifications). It was an interesting read and I appreciate the authors trying to understanding the latent structure that has been encoded in these models. However, I am just not that sure how many take-aways we can get from this study.

I am wondering how loose this categorization is , esp. for the important relations in practice. I’d be also interested in seeing more results on Freebase (and possibly Wikidata) as those KG embeddings are usually more useful. As indicated in the Appendix, the paper mentiosns most of FB15k-237 datasets are in type C, so I am just not sure how many R/S relations are actually there.

Also, according to Table 3 and Table 4, I am not sure if there are any surprising findings from there. It seems that there is some randomness/noise, but MuRE generally works better than the others. It is true that DistMult works well on the R-type relations but it is not consistent between WN and NELL.

It’d be useful to show results on more relations (and aggregated results in each category).

It'd be really great if the paper actually provides some insights on we can further improve these entity embeddings according to this categorization.


**Experience Assessment:**

I have published one or two papers in this area.

**Review Assessment: Checking Correctness Of Derivations And Theory:**

I assessed the sensibility of the derivations and theory.

**Review Assessment: Checking Correctness Of Experiments:**

I assessed the sensibility of the experiments.

**Review Assessment: Thoroughness In Paper Reading:**

I read the paper at least twice and used my best judgement in assessing the paper.

---

> ### Author Response · Authors · 2019-11-08
> **Author response to Reviewer #1: clarifications**
>
> Thank you for your review and the time taken for it, below we address each of your points in turn.
>
> * Re “take-aways”: we agree that we have not made this sufficiently clear and have updated the paper accordingly (in particular introduction, part of results and conclusion). To summarise, the key take-aways from our work are:
>   1) that the previous understanding of how semantic relations are encoded between PMI-based word embeddings for a few relations (e.g. similarity, analogies, etc - Allen & Hospedales (2019), Allen et al. (2019)) is extended to derive the difference between word embeddings for the general relations of knowledge graphs, which translate into linear algebraic mappings. From their mappings, relations can be categorised into 3 types and components of the mappings (e.g. projection matrix, translation vector) related to meaningful/interpretable semantic aspects of the relation (e.g. relatedness between entities, entity-specific features).
>   2) that PMI-based word embeddings and knowledge graph entity embeddings show commonality to their latent structure - despite the significant differences between their training data and methodology. We demonstrate this by: (i) deriving properties of the relation mappings  (based on word embeddings), e.g. vector norm, matrix symmetry/effective rank, and identifying those in actual knowledge graph representations; and (ii) showing that the relative performance of knowledge graph models for each relation type accords with how well a model’s architecture satisfies the corresponding relation conditions (based on word embeddings).
>   3) that stand-alone classification performance should be evaluated for future models since the task itself may be of more practical use than ranking metrics, and it provides novel insight into model performance.
>
> Overall, we provide an important step towards a theoretical understanding of the latent structure of knowledge graph representations. In terms of practical use, our results: provide understanding as to which model is most appropriate for a new dataset (e.g. if relations were known, a priori, to be symmetric); suggest that different aspects of relations (e.g. type, strength of relatedness) could be quantitatively evaluated; and indicate where future research effort might be directed (e.g. type C relations).
>
> * Re "FB15k-237 dataset" (FB): The prevalence of type C relations in FB does not contradict or weaken our results, it means only that FB is less useful for demonstrating the differences between model performance across relation types relative to datasets that contain a broader spread of relations by type (e.g. WN, NELL). We agree that this is insufficiently clear and move the explanation from Appx B to Sec 3.1.
>
> * Re “Table 3 & 4 findings”: we agree these are insufficiently clear and have updated the paper (Sec 3.2, to more clearly motivate the experiments, and Sec 4). Key findings of Tables 3 and 4 are:
> (i) that MuRE’s advantage over other models largely corresponds to type S/C relations, fitting prediction P1 since those relations require both additive and multiplicative components of the loss function (no other model has both);
> (ii) that, between MuRE_I and DistMult, multiplicative-only DistMult (typically) performs better for type R relations, which require a multiplicative component only; whereas additive-only MuRE_I performs best for type C/S relations (see Table 3), which require an additive component (but may also require a multiplicative component, explaining the inconsistency in Table 4); and
> (iii) performance of TuckER is comparable to MuRE_I/DistMult models for datasets with few relations (e.g. WN), but is more comparable to MuRE for datasets with many relations (e.g. NELL, FB) when multi-task learning of relations provides material benefit (as discussed in the TuckER paper, Balazevic et al.(2019b)).
> All findings accord with prediction P1 made based on the latent structure of PMI-based word embeddings (relation conditions). As a corollary, whilst multiplicative and additive link prediction models have historically jostled for superiority and which is better remained an open question, we justify why the answer is both (e.g. as in MuRE).
>
> * Re “DistMult on type R relations”: DistMult outperforms other models for type R relations, except for the “also_see” (AS) and “derivationally_related_form” (DRF) relations (Table 3).
>  - AS has a high Krackhardt score and path length indicating a tree structure and obscuring results since the models considered are not suited to hierarchical relations (as discussed in the MuRE paper, Balazevic et al (2019a)).
>  - DistMult can be fully expressed by MuRE and thus only outperforms when MuRE’s additional parameters may allow overfitting. DRF has an abundance of data (34% of all training examples), whereby any overfitting is reduced, explaining why DistMult does not outperform MuRE.

---

> ### Author Response · Authors · 2019-11-08
> **Author response to Reviewer #1: suggestions**
>
> * Re “aggregated results”: below we summarise results by relation type and model, supporting our conclusions more succinctly:
> WN18RR
>       Tr_E    M_I   Dist   Tuck  MuRE
> R    0.91   0.95   0.95   0.95   0.96
> S    0.04   0.23   0.22   0.23   0.30
> C    0.11   0.37   0.33   0.37   0.42
>
> NELL
>       Tr_E    M_I   Dist   Tuck  MuRE
> R    0.68   0.77   0.84   0.82   0.81
> S    0.37   0.51   0.58   0.61   0.64
> C    0.39   0.48   0.49   0.50   0.53
>
> * Re “insights for embedding improvement”: we provide the first theoretical insight into how relations of knowledge graphs are represented by: deriving relation-specific mappings for PMI-based word embeddings; showing that their properties are reflected in actual knowledge graph representations; and that the better a model’s architecture accommodates them, the better its performance. We also identify performance at answering standalone knowledge base queries, ie as a classifier as opposed to a ranking mechanism, giving novel insight.
> We believe that our results will enable future development of improved knowledge base representation based on a more principled understanding and by specifically identifying where to focus effort, e.g. type C relations. Combined with other works, current practitioners effectively have a “decision tree” for deciding which model to use for a new dataset depending on its properties, e.g. many relations => TuckER, highly symmetric => DistMult, hierarchical => MuRP, and in the general case MuRE (a future research direction would be to combine these). Further, our results suggest that relation properties can be identified from representation components and/or the relative performance on different models.

---

### Public Comment · ~Apoorv_Umang_Saxena1 · 2019-10-11
**Performance of TransE on R type relations**

In section 4.1, you say

[Additive models] achieve their best results on type R relations, where the relation vector can be zero/small.

However, if relation vector is 0, then model should not be able to give the correct prediction? Since head = tail if r is 0 vector. Shouldn't this be an anomaly rather than an explanation?

---

> ### Author Response · Authors · 2019-10-15
> **Type R (highly related) relations = close embeddings**
>
> Hi, thanks for your interest.
>
> Words associated by type R relations are highly "related" (in the extreme they are "similar"), i.e. they co-occur with many words in common (Sec 3). As such, their PMI vectors (and also their word embeddings) have a significant common component and so tend to have a relatively small difference (i.e. they are close). Additive-only models with a small relation vector can identify object and subject embeddings that are close, but cannot identify a relation-specific common subspace component, if required (as possible with multiplicative models). As such, additive-only models might be expected to be insufficiently discriminating. This is in fact confirmed in Table 6: additive-only model M_I predicts a high number of triples to be true for type R relations, but many of those (~69%) are in fact false (Sec 4.2).

---

> > ### Public Comment · ~Apoorv_Umang_Saxena1 · 2019-10-18
> > **Type R relation = Close embedding, evaluation**
> >
> > Hi, thanks for your answer
> >
> > If your hypothesis is that R-type relations have similar subject and object embeddings - which is your analogy to word embeddings - then shouldn't it be easier to show that by just measuring the norm of the relation vector/matrix?
> >
> > Edit: Never mind, I didn't read the full paper. The answer to this question is already there :). Sorry for bothering

---

### Author Response · Authors · 2019-11-14
**Final comments**

Dear reviewers, we thank you once again for taking the time to review our paper. We hope that we have addressed all concerns raised, in particular highlighting what our work achieves in respect of better understanding the representations learned by knowledge graph models. We would be very happy to answer any further questions you may have.

Best regards,
Authors

---

### Decision · Program_Chairs · 2019-12-19

**Decision:**

Reject

**Comment:**

The paper proposes a set of conditions that enable a mapping from word embeddings to relation embeddings in knowledge graphs. Then, using recent results about pointwise mutual information word embeddings, the paper provides insights to the latent space of relations, enabling a categorization of relations of entities in a knowledge graph. Empirical experiments on recent knowledge graph models (TransE, DistMult, TuckER and MuRE) are interpreted in light of the predictions coming from the proposed set of conditions.

The authors responded to reviewer comments well, providing significant updates during the discussion period. Unfortunately, the reviewers did not engage further after their original reviews, and so it is hard to tell whether they agreed that the changes resolved all their questions.

Overall, the paper provides much needed analysis for understanding of the latent space of relations on knowledge graphs. Unfortunately, the original submission did not clearly present the ideas, and it is unclear whether the updated version addresses all the concerns. The paper in its current state is therefore not yet suitable for publication at ICLR.

---

> ### Author Response · Authors · 2020-01-20
> **TL;DR: Rejected for lack of reviewer response**
>
> We are very disappointed that our work was rejected by the AC despite the reviewers' acceptance, essentially due to reviewers not engaging further. We accept there were points to clarify, a key purpose of the rebuttal phase, but rejecting our paper (deemed to provide "much needed analysis") due to lack of further reviewer engagement seems unreasonable and based on process rather than the paper's merit.
>
> [Even if this were a valid justification, it is incorrectly applied since Reviewer 1 did increase their score following the rebuttal, which may have gone unnoticed since they edited their initial response (see date-stamp).]
>
> We appreciate the difficult job of reviewing panels, but this is very disappointing.